# Evaluation of Criminal Sanctions Concerning Violations of Cattle and Pig Welfare

**DOI:** 10.3390/ani10040715

**Published:** 2020-04-19

**Authors:** Sofia Väärikkälä, Tarja Koskela, Laura Hänninen, Mari Nevas

**Affiliations:** 1Department of Food Hygiene and Environmental Health, Faculty of Veterinary Medicine, University of Helsinki, P.O. Box 66 (Agnes Sjöbergin katu 2), 00014 Helsingin Yliopisto, Finland; 2Research Centre for Animal Welfare, Faculty of Veterinary Medicine, University of Helsinki, P.O. Box 57, 00014 Helsingin Yliopisto, Finland; 3Department of Law, Faculty of Social Sciences and Business Studies, University of Eastern Finland, P.O. Box 111 (Yliopistokatu 2), 80101 Joensuu, Finland; 4Department of Production Animal Medicine, Faculty of Veterinary Medicine, University of Helsinki, P.O. Box 57 (Viikintie 49), 00014 Helsingin Yliopisto, Finland

**Keywords:** animal welfare crime, cattle welfare, criminal sanction, pig welfare

## Abstract

**Simple Summary:**

According to EU legislation, violations involving farm animal welfare need to be sanctioned. With this study, our aim was to assess criminal sanctions by Finnish district courts concerning cattle and pig welfare crimes. We found that small cattle farms were over-represented in court statistics. The most common sanctioned violations of cattle and pig welfare were the lack of cleanliness of premises and animals and inadequate feeding and watering. The violations were shown to have been continuing for a long time before a defendant stood trial for their actions. The penalties were lenient, i.e., the violations were not perceived as particularly blameworthy by the court. Veterinarians have a major role in initiating criminal procedures, collecting evidence, and evaluating the relevance of violations. Therefore, to improve the efficacy of criminal procedures, the police and prosecutors should work in close collaboration with veterinarians.

**Abstract:**

EU legislation requires the violations of animal welfare standards to be sanctioned. Our aim was to evaluate criminal sanctions concerning violations of cattle and pig welfare on Finnish farms. We analyzed 196 court cases heard in Finnish district courts from 2011 to 2016. Almost all the cases (95%) concerned the violations of cattle welfare, of which 61% occurred on small farms. The lack of cleanliness and inadequate feeding and watering were the most common reported violations. Median time span from the start date of an offending to a judgement was slightly less than two years. Of the cases, 96% resulted in conviction. The court did not perceive the violations as being highly blameworthy as a small fine and a short conditional imprisonment were the most often imposed sanctions. A ban on the keeping of animals was used as a precautionary measure in half of the cases. Veterinarians were shown to have an important role in the initiation of criminal procedures, providing evidence for the police, and acting as witnesses. Therefore, it is crucial to achieve a well-functioning collaboration between veterinarians and the police and prosecutors. The expertise of these authorities on animal welfare legislation should also be emphasized to improve the efficacy of criminal procedures.

## 1. Introduction

The aim of European Union (EU) animal welfare standards is to protect animals from unnecessary pain, suffering, and harm by adjusting people’s obligations and introducing prohibitions. To strengthen the enforcement, a demand for effective, proportionate, and dissuasive sanctions applicable to violations of animal welfare standards has been imposed to Member States [1]. A financial sanction may be a consequence of not complying with animal welfare standards through cross compliance [2], which links farmers’ income support provided by the EU with the compliance of certain set of animal welfare requirements. The criminal sanctions depend on the legislation of each Member State.

In Finland, animal welfare authorities are required to notify suspicions of violations of animal welfare standards to the police [3]. If there is a reason to suspect that a crime has been committed, the police will conduct a criminal investigation. In minor cases, the police may impose a fine on the defendant directly or after being confirmed by a prosecutor. These cases do not proceed to court unless the person fined refers the case to a district court for consideration. In more severe cases, depending on the outcome of the criminal investigation, a prosecutor decides whether to charge or not. If charges are brought, the case is heard in a district court, the burden of proof being on the prosecutor. The judgement is handed down by a judge, and if they are not satisfied with it, the prosecutor and/or defendant can appeal the case to a court of appeal. To be further considered by a court of appeal, the appellant needs permission, with few exceptions [4].

About 150 animal welfare offenders are convicted annually in Finland [5]. There are four animal welfare crime classes [3,6]: animal welfare infringement, petty animal welfare offence, animal welfare offence, and aggravated animal welfare offence. Animal welfare crimes can be judged as aggravated if they are particularly brutal or cruel, involve a large number of animals, or have been committed for financial purposes and if the offence is aggravated when assessed as a whole. An animal welfare crime is punishable with a fine or imprisonment. In a district court, the fine is imposed as day-fines, i.e., the amount of the day-fines depends on the income of a person sentenced. The maximum penalty for an animal welfare offence is imprisonment for two years and for aggravated animal welfare offence imprisonment for at least four months, up to four years. A sentence of up to two years of imprisonment may be imposed as conditional imprisonment with probation. [6] A person can also be banned from the keeping of animals for a fixed period or permanently. A ban should be principally imposed in all aggravated animal welfare offences; in other offence types it is discretionary. A ban may be imposed permanently if the person is convicted of an aggravated animal welfare offence, if an earlier ban on the keeping of animals had been imposed for a fixed period, or if health of the person is poor and they are permanently unfit or unable to own, keep, or care for animals [6]. The ban is a precautionary measure to prevent a person from committing a new animal welfare crime and to protect animals from further suffering [7]. The person banned from the keeping of animals may not own, keep, or care for animals or otherwise be responsible for the welfare of animals [6]. 

Violations of farm animal welfare are not rare [8,9,10], but there have been only a few studies focusing on sanctions. Arluke and Luke [11] showed that less than half of the accused animal abusers in the USA are found guilty. Regarding Finnish animal welfare crimes from 2006 to 2009, it was argued that penalties are lenient [12]. Morton et al. [13] showed that in South Australia, penalties for animal welfare offences concerning farm animals are harsher than penalties concerning companion animals. Animal abusers are a heterogenous group [14], but men are more likely to be convicted for animal welfare crimes than women [11,12]. Financial and psychiatric problems are risk factors in committing an animal welfare crime [14,15].

The focus in this study is on criminal sanctions concerning the violations of farm animal welfare. The aims of the study were (i) to analyze violations of cattle and pig welfare heard in Finnish district courts in 2011–2016, (ii) to examine the responses of animal welfare offenders to accusations, and (iii) to analyze the penalties of animal welfare crimes. Results can be utilized in the development of the criminal justice system and improving the efficacy of animal welfare control.

## 2. Materials and Methods

We requested judgements from 2011 to 2016 concerning violations of cattle and pig welfare on farms from Finnish district courts after the Legal Register Centre had granted permission to obtain the information. Further, the University of Helsinki Ethical Review Board in the Humanities and Social and Behavioural Sciences provided an ethical statement (11/2020) on the study. Cattle and pigs were chosen as these farm animals are known to be neglected most in quantity [12]. A total of 196 judgements were obtained for evaluation in this study. If a judgement given by a district court had been appealed against, the judgement of the court of appeal was requested. Pursuant to converting the judgements into an electronic format, identification data were removed. 

For the analyses of criminal procedure, we collected data on the type of a crime, the period between the start of an offending defined by a court (usually the date of the first animal welfare inspection) and institution of proceedings in a district court, and the period a case was pending legal actions in a district court. In addition, we collected information on prosecutors’ witnesses, the number of cases prosecutors had prosecuted during the study period, and the number of bans on the keeping of animals prosecutors had requested. When available from the judgement, offenders’ previous animal welfare crime history were also collected. The answers of defendants to accusations were analyzed using content analysis [16] to identify common justifications for an offending and were in general divided into three categories: confession, part confession, and denial. 

To analyze the violations of cattle and pig welfare and related farm demographic information, we collected the following data: the number and species of animals on a farm, types of violations, and duration of an offending confirmed by a court. The farms were categorized into small (fewer than 50 cattle or fewer than 250 pigs), medium (50–100 cattle or 250–750 pigs), or large (more than 100 cattle or 750 pigs). 

In the analysis of penalties, only judgements concerning single animal welfare crime were used. Only one, the harshest, penalty per case was used in the analysis. 

We used nonparametric tests as the Kolmogorov–Smirnov test (*p* < 0.05) showed data to be not normally distributed. We used the Mann–Whitney U test to make comparisons between offenders who denied their crime and those who confessed them in the duration of an offending and in the prevalence of violations and to examine the impact of a violation to the amount of the day-fines and length of an imprisonment. Kruskal–Wallis U test was used to compare the prevalence of violations between different herd sizes. We used a Chi square test with crosstabs to test relationships between violations and death animals found on a farm, urgent measures executed on a farm, an imprisonment sentence, and a ban on the keeping of animals. We used Spearman’s rank correlation coefficient to evaluate correlations between the number of inspections performed on a farm, the duration of an offending, and the harshness of penalty. All the analyses were conducted using SPSS statistical software (IBM SPSS Statistics for Windows, Version 22.0. Armonk, NY, USA: IBM Corp.). Statistical significance was accepted at a confidence level of 95% (*p* < 0.05).

## 3. Results

From the total of 196 cases concerning the violations of cattle and pig welfare on farms in Finnish district courts during 2011–2016, the median number per year was 33 (range 22–40). Of the cases, 96% (*n* = 189) resulted in conviction. Charges were dismissed in six cases (3%), and one case (1%) was withdrawn by a prosecutor due to inconsistent evidence. Of the cases, 14% (*n* = 27) had been heard by the courts of appeal. 

In 84% of the cases (*n* = 165) there was only one defendant, while two defendants were involved in 14% (*n* = 27) of the cases and three defendants in 2% (*n* = 3) of the cases, resulting in 228 defendants altogether. Of the defendants, 96% (*n* = 218) were found guilty for animal welfare crime(s), while 4% (*n* = 10) were acquitted. Of the offenders, 89% (*n* = 195) were convicted for only one animal welfare crime, while the rest were convicted for several animal welfare crimes or crime(s) other than an animal welfare crime, such as causing danger of the spread of a veterinary disease, marking and registration offences, waste offences, environmental offences, and imperilment. 

### 3.1. Criminal Procedure

Criminal procedure originated from an animal welfare inspection in most cases (98%; *n* = 192), i.e., the initiator of a procedure was an official veterinarian. Other initiators were a neighbor, a vacation substitute, and a carcasses driver. Median time span from the start date of an offending to the institution of proceedings in a district court was 1.5 years (range from 63 days to 7.5 years). The median period a case was pending in a district court was 86 days (range 14–494). 

The cases were prosecuted by 102 prosecutors. Two-thirds of the prosecutors (*n* = 68) prosecuted only one or two cases during the study period, while one-third (*n* = 34) were involved in more than two cases. The prosecutors demanded a ban on the keeping of animals in 70% (*n* = 137) of the cases. Nearly two-thirds of the demands (*n* = 91) were accepted. The prosecutors based their cases on a median of four animal welfare inspections (Table 1) and used veterinarians as witnesses in 82% (*n* = 160) of the cases. 

### 3.2. Violations and Related Farm Demographic Information (N = 189)

Of the cases resulting in conviction, 92% (*n* = 173) concerned the violations of cattle welfare, 5% (*n* = 9) of pig welfare, and 4% (*n* = 7) of both. Other animal species were involved in 16% (*n* = 30) of the cases. The number of animals on a farm during an offending was available for 74% (*n* = 140) of the cases, being on median 35 (range 1–300) and 66 (range 1–850) for cattle and pig farms, respectively. Of the cattle cases, 61% (*n* = 79) concerned small farms, while the percentages for medium sized and large farms were 21% (*n* = 27) and 19% (*n* = 24), respectively; the difference being statistically significant (Kruskal–Wallis test, *p* = 0.006). Of the pig cases, 80% (*n* = 8) concerned small farms, while the percentages for medium sized and large farms were 10% (*n* = 1) and 10% (*n* = 1), respectively. 

Median duration of an offending defined by a court was 210 days (range 1–1744, Table 1). The more animal welfare inspections had been performed on a farm, the longer an offending had lasted (Spearman’s rank *r* = 0.53, *p* < 0.001). Median time span from a start date of an offending to a conviction was 22 months (range from 3.5 months to eight years two months). The most common violations of both cattle and pig welfare were the lack of cleanliness of premises and/or animals, inadequate feeding, and inadequate watering. These violations were reported in more than half of the cases (Figure 1). Feeding was considered inadequate mainly because of lack or insufficient amount of feed (77%; *n* = 86), unsuitable feed (18%; *n* = 20), and/ or contaminated feed (14%; *n* = 16). Watering was considered inadequate mainly because of lack or insufficient amount of water (54%; *n* = 60), insufficient number of drinking places (29%; *n* = 32), dirty water (23%; *n* = 26), and/or frozen water (15%; *n* = 17). In connection with the presentation of evidence, some emphasis on the negative affective experiences of animals was put on 23% (*n* = 39), 38% (*n* = 42), and 22% (*n* = 24) of cases concerning dirtiness (pain and/or suffering due to dags or manure covering on skin), inadequate feeding (hunger), and inadequate watering (thirst), respectively. There was no statistically significant difference in the prevalence of violations between different herd sizes (Kruskal–Wallis U-test *p* > 0.05 for all). 

Dead animals due to negligence were reported in 19% (*n* = 38) of the cases. Urgent measures, e.g., euthanasia or transport of animals to a slaughterhouse, executed by an animal welfare authority were reported in 30% (*n* = 58) of the cases. Urgent measures had been executed more often if there were sick or injured animals or dead animals on a farm (49% vs. 17% and 47% vs. 26%, respectively, χ^2^-test, *p* < 0.02 for both). 

### 3.3. Responses of the Offenders (N = 218)

Half of the offenders (52%; *n* = 113) denied their guilt, while the rest confessed all (18%; *n* = 39) or some (30%; *n* = 66) charges against them. Unsafe premises, inadequate watering, and inadequate weather protection were violations that were more likely to be denied (35% vs. 16%, 43% vs. 14%, and 34% vs. 11%, respectively, χ^2^-test, *p* < 0.02 for all), whereas cases involving dead animals were confessed (48% vs. 22% *p* = 0.01). The number of inspections was higher and the duration of an offending longer in cases when an offender denied their guilt (5 vs. 3 and 403 vs. 240, Mann–Whitney U test *p* < 0.01 for both). 

A justification for an offending was given by 46% (*n* = 101) of the offenders. The three most common justifications were offender’s own health problems (56%; *n* = 57), economic problems (16%; *n* = 16), and too many animals (16%; *n* = 16).

### 3.4. Penalties

A total of 195 defendants in 172 cases were judged guilty of one animal welfare crime. A fine was the most common penalty for other than aggravated animal welfare offences (Table 2), and the median amount of the day-fines was 48 day-fines (range 15–100). The fine was followed by a conditional imprisonment with probation, and the median period for the conditional imprisonment was 90 days (Table 2). An ancillary fine, on median 43 day-fines (range 35–60), was included in every tenth (10%; *n* = 5) conditional imprisonment penalty. The rarest penalty was unconditional imprisonment (Table 2). Of the cases, 3% (*n* = 5) resulted in conviction with no penalty, i.e., the offenders were found guilty for an animal welfare crime but were not punished. Reasons for not punishing were the poor health of an offender, the death of spouse being in charge on a farm, and the efforts to improve the situation on a farm. 

The amount of the day-fines was higher in animal welfare offences than in animal welfare infringements and petty animal welfare offences (Mann–Whitney U-test *p* < 0.01 for both, Table 2). The amount of the day-fines and the length of imprisonment did not correlate with the number of inspections nor with the duration of an offending (Spearman’s rank *p* > 0.05 for both). The amount of the day-fines was higher if an offender was found guilty of inadequate feeding (54 vs. 45, Mann–Whitney U-test *p* = 0.03) or urgent measures had been executed on a farm (60 vs. 46, *p* = 0.004). A term of imprisonment was more likely if (i) an offender had a previous animal welfare conviction (69% vs. 26%, χ^2^ -test *p* = 0.001), (ii) an offender was found guilty of inadequate feeding (46% vs. 13%, *p* < 0.001), (iii) an offender was found guilty of inadequate watering (41% vs. 16%, *p* = 0.006), or (iv) urgent measures had been executed on a farm (53% vs. 20%, *p* < 0.001). An imprisonment was statistically significantly longer if (i) an offender was found guilty of inadequate watering (120 vs. 87, Mann–Whitney U-test *p* = 0.04), (ii) dead animals had been found on a farm (149 vs. 88, *p* = 0.005), or (iii) urgent measures had been executed on a farm (149 vs. 91, *p* = 0.02). A previous animal welfare conviction preceded three out of four unconditional imprisonments. Convicting an offence as aggravated was more likely if an offender had been found guilty of inadequate feeding (5% vs. 0%, χ^2^ -test *p* = 0.04) or urgent measures had been executed on a farm (10% vs. 0%, *p* = 0.003). 

In 48% (*n* = 82) of the cases at least one person was banned from the keeping of animals. The ban was more likely if dead animals had been found or urgent measures had been executed on a farm (74% vs. 42% and 67% vs. 40%, respectively, χ^2^ -test, *p* < 0.02 for both). The bans were often imposed for a fixed period (94%; *n* = 77), and only 6% (*n* = 5) of them were permanent. Of the bans, more than one third (35%; *n* = 29) applied only to one species, namely to cattle or pigs. In 29% (*n* = 24) and 21% (*n* = 17) of the cases, the ban applied to all production animals or all animals, respectively. The ban included a forfeiture in 54% (*n* = 44) of the cases: in 11 cases the animals were forfeited to state directly, and in 33 cases, the owners were reserved with the opportunity to sell animals before the forfeiture. None of the other cases included any mention of forfeiture.

## 4. Discussion

The evaluation of criminal sanctions imposed by Finnish courts revealed that penalties for farm animal welfare crimes tended to be lenient. The most often imposed penalty for other than aggravated animal offences was a fine; the mildest form of sanction in Finland. The fines imposed were at the lower end of a penal scale as the median fine was a 48 day-fine and the statutory minimum and maximum being 1 and 120 day-fines, respectively [6]. Accordingly, with a few exceptions, imprisonment sentences were short conditional imprisonments. The median length of a conditional imprisonment was 3 months for an animal welfare offence and 10 months for an aggravated animal welfare offence. The statutory minimum is 14 days and the maximum is 2 years for an animal welfare offence and 4 years for an aggravated animal welfare offence [6]. Our finding on the leniency of penalties is in line with previous studies in Finland and USA [11,12]. We argue that the court does not perceive the violations of farm animal welfare as being very blameworthy, because the sanctions were mild despite the nature and extended duration of violations.

Half of the offenders were banned from the keeping of animals. For farmers, this precautionary measure is a much more powerful sanction than a fine or conditional imprisonment as it usually means the end of their livestock farming. Permanent bans on the keeping of animals were uncommon. However, a fixed period ban may also cause a permanent end for keeping livestock as obtaining new animals after the ban may be difficult for financial reasons and because of the reluctance of other farmers to sell animals to a sentenced person. The Criminal Code of Finland [6] requires that the animals referred to in the ban on the keeping of animals should be forfeited to State. Despite that, only half of the imposed bans included a forfeiture. The rest included no mention of it, thus leaving it unclear what has happened to the animals. We argue that the forfeiture should routinely be requested by a prosecutor and it should be monitored so that the ban on the keeping of animals is carried out in practice.

The rare cases that were convicted as aggravated were shown to be associated with negligence in feeding and urgent measures executed on a farm by animal welfare authorities. The crimes concerning cattle and pigs are more likely to be about neglect than abuse. However, they are often directed at a large number of animals. In a study concerning aggravated animal welfare offences in Finland, Koskela [17] did not find any number that could be considered to be a ‘large number’. Considering financial benefit, an owner may profit when not following animal welfare standards, i.e., by not asking a veterinarian to treat sick or injured animals, by not investing in bedding material, or by not taking care of the maintenance of premises. It appears that financial benefit gained via neglecting animal welfare standards had not been acknowledged. We suggest that in the future, the financial benefit should be kept in mind when a criminal investigation is conducted. 

Veterinarians working in the field of animal welfare control were shown to have multiple important roles in a criminal procedure concerning farm animal welfare cases: they initiate the procedure by making notifications to the police, they collect evidence for the procedure, and act as witnesses. Finnish veterinarians have previously been criticized for their practice of performing repetitive animal welfare inspections without efficient enforcement measures [18]. We also show here that in many cases violations continued despite the intervention of a veterinarian and it took years, and more than three animal welfare inspections, before a defendant stood trial for their actions. Veterinarians may be uncertain about executing efficient enforcement measures when working alone [19] or notifications to the police are not done at an early stage, trusting that there will be improvements following an inspection and not wanting to place strain on an already slow crime system or aiming to collect enough evidence for the needs of the police and a prosecutor. The risk of either could lead to repetitive inspections instead of a conclusive solution. We suggest that even more efficient use of enforcement measures by veterinarians should be encouraged to protect animals from further suffering and to highlight the significance of a matter. Repetitive inspections without progress may give a distorted impression about the significance of animal welfare violations, not only on the target of control but also on the police, prosecutor, and judge. We showed that the number of inspections did not correlate with the harshness of the penalty, whereas a harsher penalty and a ban on the keeping of animals were more likely if urgent measures had been executed by the animal welfare authority.

We found that only 4% of the cases heard in the district court resulted in a not guilty verdict. In comparison, the percentage was 16% in South Australia [13]. The low prevalence of acquittals may indicate that veterinarians initiate a criminal procedure only in severe cases, or that the police conduct a preliminary investigation, or a prosecutor prosecutes only if they perceive that evidence is strong enough. More research is needed to explain the low prevalence of acquittals. We agree with Touroo and Fitch [20] that official veterinarians should have the competence to identify, collect, and preserve evidence and a good understanding of the animal welfare standards that are applicable. Animal welfare cases should also be investigated and prosecuted more eagerly, and to achieve this, the police and the prosecutors should be familiar with animal welfare legislation and have experience of animal welfare cases. We agree with Koskela [21] that animal welfare cases should be concentrated to certain police officers and prosecutors. The long gap before a defendant stood trial for their actions is problematic from the perspective of animal welfare, as animals are usually cared for by a defendant and there is a risk that violations continue during the pending time. Additionally, a defendant may not understand the handling of old matters as situation on a farm evolves all the time.

Harsher penalties were imposed if feeding or watering had been neglected or if animals had died due to a negligence. The importance of feeding and watering for the welfare of an animal is self-evident; however, the relevance of other violations may not be as comprehensible, thus leading to a lower penalty. There are many violations that significantly affect the welfare of an animal, e.g., the lack of cleanliness of cattle causes suffering in the form of irritations and skin infections, and inadequate care of sick or injured animals in the form of pain, and their inability to behave normally. Although some emphasis on negative affective experiences of animals had been presented during court hearings, we agree with Ledger and Mellor [22] that these experiences should be highlighted more in animal neglect or cruelty cases. The value of a veterinarian acting as a witness is emphasized in the assessment of suffering and other negative affective experiences. We suggest that the simplified “Five Domains model” by Ledger and Mellor [22] should be adopted by veterinarians working in the field of animal welfare control. Cooper and Cooper [23] have also recognized the importance of veterinarians in the assessment of welfare and in the determination of the circumstances of the death of an animal. The high prevalence of offenders who denied their guilt may indicate the differences of opinion over complying with animal welfare standards between farmers and animal welfare authorities and their different understanding of negative welfare outcomes to animals. The legal requirements behind the violations the offenders were often found to deny should be clarified to farmers. 

Health and economic problems were found to be the justifications given most often for animal welfare crimes. One possible reason for the high prevalence of small farms in our data is that small farms are less likely to be visited by outsiders, e.g., veterinarians and vacation substitutes, and thus human and animal welfare problems may remain hidden for longer and escalate. Having more animals than those that can be reared properly and incapability to invest economically are probably also more likely in small farms with older premises and restricted space and facilities. Andrade and Anneberg [15] and Devitt et al. [24] have emphasized the complexity of animal neglect and identified the association with health, social, and financial problems. Response to animal neglect has been criticized for being too one-dimensional; focusing purely on animals [25]. We agree that a multiagency and multidisciplinary approach would be more functional in animal neglect cases [25,26], i.e., collaboration with officials for social welfare and health care should be developed since cases often originate from human problems. 

## 5. Conclusions

We showed that cattle and pig welfare cases proceeding to a district court are most often concerned with the lack of cleanliness and the inadequate feeding and watering of animals. The criminal procedure was a time-consuming process resulting in lenient penalties. The role of veterinarians in collecting evidence and evaluating the relevance of violations was important. We conclude that the threshold for initiating a criminal procedure should be kept low and there should be additional investment to achieve well-functioning cooperation between the police, prosecutors, and official veterinarians. Furthermore, the expertise of these authorities on animal welfare legislation is crucial. 

## Figures and Tables

**Figure 1 animals-10-00715-f001:**
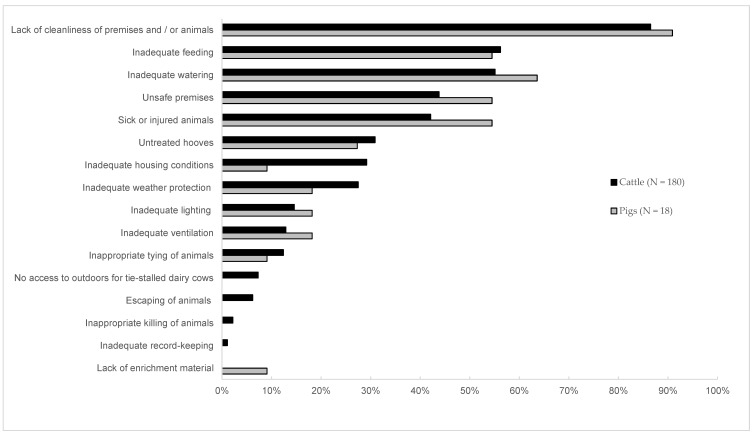
The proportion of different forms of animal welfare violations against cattle and pigs from all violations on the farm cases heard in the Finnish district courts during 2011–2016.

**Table 1 animals-10-00715-t001:** The number of animal welfare inspections used as a base for prosecution and median duration of an offending confirmed by a court in animal welfare cases (*N* = 189) heard in Finnish district courts during 2011–2016.

Inspections	*n* (%)	Duration of Offending (days)	Range
1	32 (17)	57	1–1304
2	32 (17)	136	1–1744
3	19 (10)	133	45–926
4	17 (9)	255	21–1435
5	21 (11)	314	29–907
6	15 (8)	405	13–1104
7	14 (7)	630	162–1178
8	4 (2)	456	380–1322
9	6 (3)	323	30–1069
10	5 (3)	484	79–1087
12	2 (1)	499	396–602
13	2 (1)	996	732–1259
14	4 (2)	1289	649–1642
17	1 (1)	1505	1505
NK ^1^	15 (8)	83	1–400

^1^ NK, not known. The number of inspections was not traceable from the judgements.

**Table 2 animals-10-00715-t002:** Penalties and bans on the keeping of animals per offence type for violations of cattle and pig welfare imposed by the Finnish district courts in 2011–2016.

Offence Type	n (%)	Fine n, Amount (Range) ^1^	Conditional Imprisonment n, Length (Range) ^2^	Unconditional Imprisonment n, Length (Range) ^2^	Ban n (%)	Length of Fixed Ban (Range) ^3^
Animal welfare infringement	7 (4.1%)	7, 24 (15–50)	NA	NA	0 (0)	NA
Petty animal welfare offence	4 (2.3%)	4, 28 (15–40)	NA	NA	0 (0)	NA
Animal welfare offence	156 (90.7%)	103, 51 (15–100)	45, 92 (40–365)	3, 140 (120–180) ^4^	77 (49.4)	4 (1–15) ^5^
Aggravated animal welfare offence	5 (2.9%)	NA	4, 309 (240–425)	1, 120 (120)	5 (100)	5 (5–6) ^6^

^1^ day-fines; ^2^ days; ^3^ years; ^4^ all three imprisonments were changed to community service; ^5^ in addition four permanent bans; ^6^ in addition one permanent ban; and NA, not applicable.

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
