# Peer review of "Evaluation of Criminal Sanctions Concerning Violations of Cattle and Pig Welfare"

_animals, 2020, doi:10.3390/ani10040715_

Round 1
Reviewer 1 Report
This is an extremely interesting study that reports on animal welfare infringements in the cattle and pig industries in Finland between 2011 and 2016. The authors have looked at the relationship between a number of variables relating to farm demographics, the type, severity and duration of offending, and prosecution outcomes. Overall, they conclude that the length of time taken for cases to be heard is too long and that penalties imposed are too lenient (in the context of existing legislative guidelines) – a concerning outcome.
It is interesting that you note that violations which have a less obvious impact on an animal’s welfare do not receive such harsh penalties. This is a problem that other countries have also encountered. I refer you to a publication by Dr Rebecca Ledger that may be of interest, it focusses on the issue of making legal decisions on animal welfare outcomes in the absence of physical/clinical evidence of cruelty or neglect: Ledger, R. A. and D. J. Mellor. 2018. Forensic Use of the Five Domains Model for Assessing Suffering in Cases of Animal Cruelty. Animals. doi:10.3390/ani8070101.
Some minor English editing is required e.g. the redundant use of 'the' in many instances, the use of present tense in some places when discussing results.
Specific points:
Line 45: Sentence should be amended e.g. to “…by adjusting people’s obligations and introducing prohibitions”. Otherwise this reads as adjusting people’s prohibitions, which doesn’t make sense.
Line 86: Please change “the aim of the study was” to “the aims of the study were”. Aim 1 (to assess violations from the animal welfare perspective) is not very specific or measurable. What do you mean by assess? Perhaps this should be something like ‘to analyse violations of cattle and pig welfare prosecuted in Finnish district courts between 2011 and 2016’.
Line 100: How did you ascertain the ‘known start of a crime’? Was this the date it was first recorded by the veterinarian? Please clarify.
Line 103: Please remove the word ‘been’ from this sentence
Line 108: Please rephrase “To assess the judgements from animal welfare perspective” as this suggests some form of welfare assessment, whereas you are really just collecting and analysing farm demographic information.
Lines 114–118: It would be very useful to describe in more detail here which specific relationships were investigated, as well as defining the tests used.
Line 137: Please omit averages/means when the median has been reported, unless you are referring to normally distributed continuous variables, in which case the mean (standard deviation) should be provided instead of the median (range).
Line 146: Please clarify what you mean by “median duration of animal welfare cases” in the table heading. Do you mean the duration of offending prior to being charged? This ties in with my previous comment regarding the known start time of the crime.
Line 151: In the sentence “The median number of animals on a farm during the crime became apparent in 74% (n = 151 140) of the cases, being 35…” the phrase ‘became apparent’ does not fit. Do you mean that data was available for 74% of cases? Please rephrase for clarity.
Line 158: Again, need to define what ‘duration of the crime means’ – if you define this earlier in the text, no change is needed here.
Line 179–80: Can you please clarify what you mean by defendants pleading guilty ‘totally’ or ‘partly’. Does this mean they confessed to all or only some charges against them?
Line 200: Please expand the table caption so that the table can be understood without reference to the text.
Line 203, 204, 206: Please change ‘the amount of fine’ to ‘the size of the fine’ for clarity.
Line 240: For clarity, please replace ‘continuation’ with ‘duration’.
Line 247: Needs rewording for clarity, e.g. …requires that the animals referred to in the ban on the keeping of animals should be ordered to forfeited to the State.
Lines 252–254: The sentence describing the criteria for crime to be judged ‘aggravated’ should be included in the introduction, where the animal welfare crime classes are introduced.
Line 276: In this paragraph discussing the low prevalence of acquittals, this is attributed to prosecutors only prosecuting cases where they believe a conviction will result. Does this mean that there are cases initiated by veterinarians that don’t get to trial? Or, might this be because veterinarians themselves are only initiating proceedings in more severe cases?
Line 282: What do you mean when you say that “efficient use of enforcement measures by veterinarians should be encouraged”– this seems to be suggesting that veterinarians aren’t doing as much as they could. Could you please elaborate on this?
Line 327: Please replace “their” with “veterinary” for better clarity.
This is a great piece of work, yielding some really important information. Hopefully the data you have produced will help to initiate long-term changes that will positively impact on animal welfare.
Author Response
It is interesting that you note that violations which have a less obvious impact on an animal’s welfare do not receive such harsh penalties. This is a problem that other countries have also encountered. I refer you to a publication by Dr Rebecca Ledger that may be of interest, it focusses on the issue of making legal decisions on animal welfare outcomes in the absence of physical/clinical evidence of cruelty or neglect: Ledger, R. A. and D. J. Mellor. 2018. Forensic Use of the Five Domains Model for Assessing Suffering in Cases of Animal Cruelty. Animals. doi:10.3390/ani8070101.
Lines 185-189 and 341-346: We thank you for referring us this interesting article. Inspired by the article, we studied whether negative animal welfare outcomes of the most common violations had been brought up in Finnish court hearings. We added the results in the result section “In connection with the presentation of evidence some emphasis on the negative affective experiences of animals was put on 23% (n = 39), 38% (n = 42) and 22% (n = 24) of cases concerning dirtiness (pain and/ or suffering due to dags or manure covering on skin), inadequate feeding (hunger) and inadequate watering (thirst), respectively.” and also some discussion “Although some emphasis on negative affective experiences of animals had been put during court hearings, we agree with Ledger and Mellor [22] that these experiences should be highlighted more in animal neglect or cruelty cases. The value of a veterinarian acting as a witness is emphasized in the assessment of suffering and other negative affective experiences. We suggest that the simplified “Five Domains model” by Ledger and Mellor [22] should be adopted by veterinarians working in the field of animal welfare control.”
Some minor English editing is required e.g. the redundant use of 'the' in many instances, the use of present tense in some places when discussing results.
We checked English spelling throughout the text.
Specific points:
Line 45: Sentence should be amended e.g. to “…by adjusting people’s obligations and introducing prohibitions”. Otherwise this reads as adjusting people’s prohibitions, which doesn’t make sense.
Line 45: The sentence was reworded as suggested: “…by adjusting people’s obligations and introducing prohibitions.”
Line 86: Please change “the aim of the study was” to “the aims of the study were”. Aim 1 (to assess violations from the animal welfare perspective) is not very specific or measurable. What do you mean by assess? Perhaps this should be something like ‘to analyse violations of cattle and pig welfare prosecuted in Finnish district courts between 2011 and 2016’.
Lines 89-90 : We agree that verb “analyse” is more appropriate in this context so the sentence was reworded as “The aims of the study were i) to analyse violations of cattle and pig welfare heard in Finnish district court in 2011-2016…”
Line 100: How did you ascertain the ‘known start of a crime’? Was this the date it was first recorded by the veterinarian? Please clarify.
Lines 104-105: We used the date that court had confirmed as the starting date of an offending, mostly it was the date when a farm had been inspected for the first time. This was clarified as “the period between the start of an offending defined by a court (usually the date of the first animal welfare inspection) and institution of proceedings in a district court…”. We also used the word “offending” instead of “crime”.
Line 103: Please remove the word ‘been’ from this sentence
Line 108: The word “been” was removed as suggested: “…prosecutors had requested.”
Line 108: Please rephrase “To assess the judgements from animal welfare perspective” as this suggests some form of welfare assessment, whereas you are really just collecting and analysing farm demographic information.
Lines 113-114: The sentence was reworded as “To analyse the violations of cattle and pig welfare and related farm demographic information, we collected…”
Lines 114–118: It would be very useful to describe in more detail here which specific relationships were investigated, as well as defining the tests used.
Lines 121-130: The used tests and analysis were defined more thoroughly: “We used non-parametrical tests as Kolmogorov-Smirnov test (p < 0.05) showed data to be not normally distributed. We used Mann-Whitney T test to make comparisons between offenders who denied their crime and those who confessed them in the duration of an offending and in the prevalence of violations and to examine the impact of a violation to the amount of the day-fines and length of an imprisonment. Kruskal-Wallis U test was used to compare the prevalence of violations between different herd sizes. We used Chi square test with crosstabs to test relationships between violations and death animals found on a farm, urgent measures executed on a farm, an imprisonment sentence, and a ban on the keeping of animals. We used Spearman’s rank correlation coefficient to evaluate the correlations between the number of inspections performed on a farm, the duration of an offending, and the harshness of penalty.”
Line 137: Please omit averages/means when the median has been reported, unless you are referring to normally distributed continuous variables, in which case the mean (standard deviation) should be provided instead of the median (range).
The means/averages and standard deviations were removed throughout the text since the data were not normally distributed. There was some confusion on terms and therefore all medians were re-calculated.
Line 146: Please clarify what you mean by “median duration of animal welfare cases” in the table heading. Do you mean the duration of offending prior to being charged? This ties in with my previous comment regarding the known start time of the crime.
Line 159-161: To clarify the title it was reworded as “Table 1. The number of animal welfare inspections used as a base for prosecution and median duration of an offending confirmed by a court in animal welfare cases (N = 189) heard in Finnish district courts during 2011-2016.” Standard deviations were removed from the table.
Line 151: In the sentence “The median number of animals on a farm during the crime became apparent in 74% (n = 151 140) of the cases, being 35…” the phrase ‘became apparent’ does not fit. Do you mean that data was available for 74% of cases? Please rephrase for clarity.
Lines 167-168: The sentence was reworded as: “The number of animals on a farm during an offending was available for 74% (n = 140) of the cases, being on median…”
Line 158: Again, need to define what ‘duration of the crime means’ – if you define this earlier in the text, no change is needed here.
Lines 174-175: The sentence was reworded to make it clearer: “Median duration of an offending defined by a court was 210 days…”
Line 179–80: Can you please clarify what you mean by defendants pleading guilty ‘totally’ or ‘partly’. Does this mean they confessed to all or only some charges against them?
Lines 204-205: The sentence was clarified by rewording it: “…while the rest confessed all (18%; n = 39) or some (30%; n = 66) charges against them.”
Line 200: Please expand the table caption so that the table can be understood without reference to the text.
Line 226: The table caption was reworded as “Table 2. Penalties and bans on the keeping of animals per offence type for violations of cattle and pig welfare imposed by the Finnish district courts in 2011-2016.”
Line 203, 204, 206: Please change ‘the amount of fine’ to ‘the size of the fine’ for clarity.
We would prefer using “the amount” as this term is more appropriate for day-fines according to Finlex Data Bank (official database on judicial information of Finland). However, to clarify the text we used “the amount of the day-fines” instead of “the amount of fine” throughout the text.
Line 240: For clarity, please replace ‘continuation’ with ‘duration’.
Line 268: The word “continuation” was replaced with “duration” as suggested.
Line 247: Needs rewording for clarity, e.g. …requires that the animals referred to in the ban on the keeping of animals should be ordered to forfeited to the State.
Lines 275-276: The sentence was reworded as suggested: “…requires that the animals referred to in the ban on the keeping of animals should be forfeited to State.”
Lines 252–254: The sentence describing the criteria for crime to be judged ‘aggravated’ should be included in the introduction, where the animal welfare crime classes are introduced.
Lines 63-65: The sentence describing the criteria for aggravated animal welfare offence was moved to the introduction.
Line 276: In this paragraph discussing the low prevalence of acquittals, this is attributed to prosecutors only prosecuting cases where they believe a conviction will result. Does this mean that there are cases initiated by veterinarians that don’t get to trial? Or, might this be because veterinarians themselves are only initiating proceedings in more severe cases?
Lines 312-318: This is a good question and more research is needed to explain the high prevalence of acquittals. The sentence was reworded as “We found that only four per cent of the cases heard in the district court resulted in a not guilty verdict. In comparison, the percentage was 16% in South Australia [13]. The low prevalence of acquittals may indicate that veterinarians initiate a criminal procedure only in severe cases, or that the police conduct a preliminary investigation or a prosecutor prosecutes only if they perceive that evidence is strong enough. More research is needed to explain the low prevalence of acquittals.”
Line 282: What do you mean when you say that “efficient use of enforcement measures by veterinarians should be encouraged”– this seems to be suggesting that veterinarians aren’t doing as much as they could. Could you please elaborate on this?
Lines 297-308: To clarify our argument that veterinarians do not always use enough efficient enforcement measures we moved the sentence to previous paragraph where we have discussed about the repetitive inspections. The use of enforcement measures is considered as inefficient since in some cases violations have continued years despite intervention of veterinarians. “We also show here that in many cases violations continued despite the intervention of a veterinarian and it took several years, and more than three animal welfare inspections, before a defendant stood trial for their actions. Veterinarians may be uncertain about executing efficient enforcement measures when working alone [19] or notifications to the police are not done at an early stage, trusting that there will be improvements following an inspection and not wanting to place strain on the already slow crime system or aiming to collect enough evidence for the needs of the police and a prosecutor. The risk of either could lead to repetitive inspections instead of a conclusive solution. We suggest that even more efficient use of enforcement measures by veterinarians should be encouraged to protect animals from further suffering and to highlight the significance of a matter. Repetitive inspections without progress may give a distorted impression about the significance of animal welfare violations, not only on the target of control but also on the police, prosecutor and judge.”
Line 327: Please replace “their” with “veterinary” for better clarity.
Line 377: The sentence was reworded “Also, the expertise of these authorities on animal welfare legislation is crucial.” We believe that it is also important that the police and prosecutors are familiar with animal welfare legislation.
Reviewer 2 Report
Thanks a lot for a good paper. I would suggest just three moderate changes.
L163. Taking into account that this is a journal focused on animals it would be very appreciated some more details about what type of inadequate watering (not water at all, a high number of animals per drinker, dirty water...) or inadequate feeding (animals with vey bad body condition, contaminated food, animals with no access to food…) is described in these cases.
L187. Just clarify if these health problems are related to the farmer or to the cattle or pigs
L319. Include here the argues given by the farmers... something like: "too much animals (so lack of personnel), or having more animals than those that can be reared properly, and economic reasons (incapacity for doing the same investment in the case of small farms than in the case of bigger ones).
Author Response
Dear Reviewer,
We thank you for your valuable comments on the manuscript ‘Evaluation of criminal sanctions concerning violations of cattle and pig welfare’. We have now revised the manuscript. The revised parts are marked in the manuscript and the line numbers given refer to those in the corrected manuscript.
L163. Taking into account that this is a journal focused on animals it would be very appreciated some more details about what type of inadequate watering (not water at all, a high number of animals per drinker, dirty water...) or inadequate feeding (animals with vey bad body condition, contaminated food, animals with no access to food…) is described in these cases.
Lines 181-185: We added more details of inadequate feeding and watering in the results section: “Feeding was considered inadequate mainly because of lack or insufficient amount of feed (77%; n = 86), unsuitable feed (18%; n = 20) and/ or contaminated feed (14%; n = 16). Watering was considered inadequate mainly because of lack or insufficient amount water (54%; n = 60), insufficient amount of drinking places (29%; n = 32), dirty water (23%; n = 26), and/or frozen water (15%; n = 17).”
L187. Just clarify if these health problems are related to the farmer or to the cattle or pigs
Line 213: This was clarified by stating: “…were offender’s own health problems…”
L319. Include here the argues given by the farmers... something like: "too much animals (so lack of personnel), or having more animals than those that can be reared properly, and economic reasons (incapacity for doing the same investment in the case of small farms than in the case of bigger ones).
Lines 354-359: The sentence was moved to the beginning of the paragraph: “Health and economic problems were found to be the justifications given most often for animal welfare crimes. A reason for the high prevalence of small farms in our data is that small farms are less likely to be visited by outsiders, e.g. veterinarians and vacation substitutes, and thus human and animal welfare problems may remain hidden for longer and escalate. Having more animals than those that can be reared properly, and incapability to invest economically are probably also more likely in small farms with older premises and restricted space and facilities.”